# Adaptive design methods in dialysis clinical trials: a systematic review protocol

Conor Judge [1,2,3,4] Robert P Murphy [1] Sarah Cormican,[3,4] Andrew Smyth,[1,4] Martin O'Halloran,[2] Martin O'Donnell[1]

[1]HRB-Clinical Research Facility, National University of Ireland, Galway, Co. Galway, Ireland
[2]Translational Medical Device Lab, National University of Ireland Galway, Galway, Co. Galway, Ireland
[3]Wellcome Trust – HRB, Irish Clinical Academic Training, National University of Ireland Galway, Galway, Ireland
[4]Deparrtment of Nephrology, Galway University Hospital, Galway, Ireland

**Correspondence to**
Dr Conor Judge;
conor.judge@nuigalway.ie

## ABSTRACT

**Introduction** Adaptive design methods are a potential solution to improve efficiency of clinical trials but their uptake in dialysis is unknown. We aim to investigate the use of adaptive design methods in dialysis clinical trials and to cultivate further adoption of adaptive design methods by the nephrology community.

**Methods and analysis** We will adhere to the Preferred Reporting Items for Systematic Review and Meta-Analysis Protocols guidelines and the Cochrane Collaboration Handbook. We will perform a literature search through MEDLINE (PubMed), EMBASE and CENTRAL, a detailed data extraction of trial characteristics and a narrative synthesis of the data. There will be no language restrictions. We will estimate the percentage of adaptive clinical trials per year in dialysis. Subgroup analysis will be performed by dialysis modality, funder and geographical location.

**Ethics and dissemination** Ethical approval will not be required for this study as data will be obtained from publicly available clinical trials. We will disseminate our results in a peer-reviewed publication.
PROSPERO registration number

## Strengths and limitations of this study

► This study presents a comprehensive methodology for a systematic review of the current state of adaptive designs (ADs) clinical trials in dialysis.
► Two researchers will independently perform the study selection, data extraction and risk of bias assessment.
► Detailed characteristics of trials will be collected and analysed including nature of intervention, funding source, dialysis modality and the nature of the AD.
► Subgroup analysis will be performed to highlight differences in design characteristics between dialysis modality, temporal trends, types of intervention, funding and geographical location.

## INTRODUCTION
### Background
Randomised controlled trials (RCTs) are the gold standard for confirming efficacy or futility of new therapies.[1] Nephrology as a specialty, and especially patients with end-stage renal disease (ESRD), has traditionally had a low number of randomised trials compared with similar specialities.[2] Many reasons have been cited for this including recruitment difficulty, history of underpowered trials and lack of funding.[3 4] Nephrology studies are increasing in number but still lag behind other specialities.[5]

There are an estimated 726 331 prevalent cases of ESRD in the USA[6] and globally, in 2010, an estimated 2.3–7.1 million people with end-stage kidney disease (ESKD) died without access to chronic dialysis.[7] Projections estimate a 11%–18% increase in the crude incidence rate from 2015 to 2030.[8] People with ESRD represent 1% of the Medicare population but account for 7% of Medicare's expenditures.[9] Additionally, people with ESRD experience higher mortality,[10] morbidity and worse quality of life than the general public.[11]

Adaptive clinical trials use the results from interim data analysis of ongoing trials to modify the study design in a predefined way.[12] This is performed without undermining the integrity or validity of the trial and thus preserving the type 1 error (false positive) rate. The most common type of adaptive design (AD) is the Group Sequential Trial, where planned interim analysis enables stopping of trials for efficacy or futility. Other designs included sample size re-estimation, multiarm multistage (MAMS) trials, adaptive randomisation, biomarker adaptive and seamless phase II/III trials.[13] Adaptive clinical trials would appear particularly suitable for evaluation of novel interventions in ESRD and according to the Food and Drug Administration, ADs could reduce resource requirements, decrease time to study completion and increase the likelihood of study success.[14] A specific problem in previous ESRD trials includes over reliance on observational data in the design of clinical trials.[15] Observational

data are used to inform assumptions about the expected effect size and SD of effect size for power calculations. If these assumptions are incorrect, they can result in an underpowered trial with an insufficient sample size to answer the research question. Adaptive sample size re-estimation has been used successfully in cardiology trials to help this problem.[16] Planned blinded sample size re-estimation can discover incorrect underlying assumptions and trigger increased recruitment targets mid trial to ensure adequate power to answer the research question. A second specific problem in previous ESRD trials is extrapolation of results from populations without Chronic Kidney Disease (CKD). An example of this was the 4D study,[17] where 20 mg/day of atorvastatin in patients with diabetes and ESRD did not reduce cardiovascular events, despite showing a 20%–30% reduction in populations without CKD.[18] One concern was that a single level of statin or a single low-density lipoprotein cut-off would not convey the same benefit in a ESKD population compared with a population without CKD.[15] Adaptive MAMS trials have been successfully used in HIV trials, Telmisartan and Insulin Resistance in HIV used a MAMS design with one interim analysis to access three telmisartan doses (20, 40 and 80 mg daily).[19] MAMS designs are partially suited for populations with ESRD where, compared with populations without CKD, pharmacokinetic differences can be large and alternative drug doses are often required.[20]

## Objectives

The systematic review will aim to: (1) summarise the use of AD methods in dialysis clinical trials per year, (2) describe the characteristics of the ADs including dialysis modality, funding and geographical location, (3) describe the AD characteristics of the trial, (4) estimate the percentage of adaptive clinical trials in dialysis and (5) outline trends in all the above.

## METHODS AND ANALYSIS

This systematic review has been registered with PROSPERO (registration number: 163 946 (Temporary ID)). We will conduct this systematic review according to the Preferred Reporting Items for Systematic Review and Meta-Analysis Protocols (PRISMA-P) and have completed the PRISMA-P checklist (online supplementary appendix 1).

### Inclusion/exclusion criteria for the selection of studies
#### Type of study design and participants

RCTs of interventions in patients with ESRD or acute kidney injury (AKI) undergoing renal replacement therapy (RRT) including haemodialysis, peritoneal dialysis, haemodiafiltration and haemofiltration. We will not limit our population or any specific disease.

### Type of intervention

We will not place a restriction on the intervention type and will include trials that study medications during dialysis, medical devices, dialysis parameters, and so on.

### Type of outcome

We will include all outcomes including surrogate markers, patient-centred outcomes and hard clinical outcomes.

### Search method for the identification of trials
#### Electronic search

We will perform electronic searches on MEDLINE (PubMed), EMBASE and CENTRAL from database inception until 1 January 2020. Zotero will be used as our reference manager and the Revtools package on R will be used to eliminate duplicate records. The search will be conducted in English. The dialysis search terms were adapted from Beaubien-Souligny et al[21] and include dialysis(tiab) OR peritoneal dialysis(tiab) OR hemodialysis(tiab) OR hemodiafiltration(tiab) OR haemodiafiltration(tiab) OR hemofiltration(tiab) OR haemofiltration OR extracorporeal blood cleansing(tiab) OR haemodialysis(tiab) OR Renal Dialysis(mh) OR Renal replacement(tiab) OR end stage kidney(tiab) OR end stage renal(tiab) OR stage five kidney(tiab) OR stage five renal(tiab).

The AD search terms were adapted from Bothwell et al[22] and include 'phase ii/iii'(tiab) OR 'treatment switching'(tiab) OR 'biomarker adaptive'(tiab) OR 'biomarker adaptive design'(tiab) OR 'biomarker adjusted'(tiab) OR 'adaptive hypothesis'(tiab) OR 'adaptive dose-finding'(tiab) OR 'pick-thewinner'(tiab) OR 'drop-the-loser'(tiab) OR 'sample size re-estimation'(tiab) OR 're-estimations'(tiab) OR 'adaptive randomization'(tiab) OR 'group sequential'(tiab) OR 'adaptive seamless'(tiab) OR 'adaptive design'(tiab) OR 'Interim monitoring'(tiab) OR 'Bayesian adaptive'(tiab) OR 'Flexible design'"(tiab) OR 'Adaptive trial'(tiab) OR 'play-the-winner'(tiab) OR 'adaptive method'(tiab) OR (adaptive(All Fields) AND dose(All Fields) AND adjusting(All Fields)) OR 'response adaptive'(All Fields) OR 'adaptive allocation'(All Fields) OR 'adaptive signature design'(tiab) OR 'treatment adaptive'(tiab) OR 'covariate adaptive'(tiab) OR

'sample size adjustment'(tiab). We will perform two searches, first, we will combine the dialysis and adaptive search terms using a Boolean AND. The specific search will be carried out as described in table 1. Second, we will perform a search with the dialysis search terms without the adaptive search terms for calculation of our total RCTs in dialysis denominator.

### Selection and analysis of trials

We will use the high sensitivity machine learning classifier (RobotSearch) to identify RCTs from the combined search (dialysis and AD) and from the dialysis search.[23] RobotSearch is a machine learning classification algorithm combining an ensemble of support vector machines and convolutional neural networks with a reported area under the curve of 0.987 (95% CI, 0.984 to 0.989). We

**Table 1** Search strategy for MEDLINE (PubMed)

| | | |
|---|---|---|
| dialysis[tiab] | AND | Phase ii/iii(tiab) |
| OR | | OR |
| peritoneal dialysis(tiab) | | treatment switching(tiab) |
| OR | | OR |
| hemodialysis(tiab) | | biomarker adaptive(tiab) |
| OR | | OR |
| hemodiafiltration(tiab) | | biomarker adaptive design(tiab) |
| OR | | OR |
| haemodiafiltration(tiab) | | biomarker adjusted(tiab) |
| OR | | OR |
| hemofiltration(tiab) | | adaptive hypothesis(tiab) |
| OR | | OR |
| haemofiltration | | adaptive dose-finding(tiab) |
| OR | | OR |
| extracorporeal blood cleansing(tiab) | | pick-the winner(tiab) |
| OR | | OR |
| haemodialysis(tiab) | | drop-the-loser(tiab) |
| OR | | OR |
| Renal Dialysis(mh) | | sample size re-estimation(tiab) |
| OR | | OR |
| Renal replacement(tiab) | | re-estimations(tiab) |
| OR | | OR |
| end stage kidney(tiab) | | adaptive randomization(tiab) |
| OR | | OR |
| end stage renal(tiab) | | group sequential(tiab) |
| OR | | OR |
| stage five kidney(tiab) | | adaptive seamless(tiab) |
| OR | | OR |
| stage five renal(tiab) | | adaptive design(tiab) |
| | | OR |
| | | Interim monitoring(tiab) |
| | | OR |
| | | Bayesian adaptive(tiab) |
| | | OR |
| | | Flexible design(tiab) |
| | | OR |
| | | Adaptive trial(tiab) |
| | | OR |
| | | play-the-winner(tiab) |
| | | OR |
| | | adaptive method(tiab) |
| | | OR |
| | | (adaptive(All Fields) AND dose(All Fields) AND adjusting(All Fields)) |
| | | OR |
| | | response adaptive(All Fields) |

Continued

**Table 1** Continued

| | |
|---|---|
| | OR |
| | adaptive allocation(All Fields) |
| | OR |
| | adaptive signature design(tiab) |
| | OR |
| | treatment adaptive(tiab) |
| | OR |
| | covariate adaptive(tiab) |
| | OR |
| | sample size adjustment(tiab). |

will manually confirm a random sample (10%) of studies classified as 'Not RCT' after the RobotSearch screening step. We will then review the title and abstracts of studies to confirm that they are RCTs and identify trials with AD methods for inclusion or exclusion. Studies with insufficient information to determine use of AD methods will also be included for full-text review. We will then perform full-text review to confirm studies that will be included in the final systematic review. This process will be summarised in a PRISMA flowchart. Abstract, title and full-text review will be performed by CJ and RPPM. Disagreements will be resolved by consensus or by a third reviewer (SC), if necessary.

CJ and RPPM will extract the following information (adapted from Hatfield *et al* 2016)[24] in parallel and record in a custom database (summarised in table 2):

**Table 2** Characteristics of the trials

| Study characteristics | Categories | Description |
|---|---|---|
| Nature of AD | Group Sequential Design (GSD) / Sample Size Re-estimation (SSR) / Dose Selection (DS)/ Dose Escalation (DE) / Seamless / Interim Analysis | The type of AD used in the trial. |
| Stopping rule | Futility/Efficacy/Two sided/N/A | If a stopping rule was used, what was the nature of the stopping rule. |
| Year of study completion | None | The year of study completion. |
| Population under study | None | A description of the population studied for example, patients with diabetes. |
| Chronicity of RRT | AKI/ESKD | A category for the chronicity of RRT, either AKI or ESRD. |
| Intervention | None | A free-text description of the intervention. |
| Nature of the intervention | Medication/Medical Device/Dialysis Parameter | A category for the nature of the intervention. |
| Primary outcome | None | A description of the primary outcome of the trial. |
| Type of primary outcome | Continuous or dichotomous | A categorial variable for the type of primary outcome variable. |
| Nature of primary outcome | Surrogate, patient-centred or hard clinical | A categorial variable for the nature of primary outcome variable either surrogate, patient-centred or hard clinical. |
| Dialysis modality | Haemodialysis, peritoneal dialysis, haemodiafiltration or haemofiltration | A categorial variable for the dialysis modality. |
| Sample size of study | None | The number of participants in the study. |
| The country of the lead investigator | None | The country of the lead investigator. |
| The funder of the study | Public/Private | A categorial variable for source of funding for the study. |
| Study phase | Phase II/Phase III/Combined Phase II/III | A categorial variable for study phase. |

AD, adaptive design; AKI, acute kidney injury; ESKD, end-stage kidney disease; ESRD, end-stage renal disease; RRT, renal replacement therapy.

1. The type of the AD for example, dose-finding, adaptive hypothesis, group sequential, adaptive randomisation, seamless phase II/III, adaptive treatment-switching, biomarker adaptive and sample size re-estimation.
2. Stopping rule (futility or efficacy).
3. Year of completion of study.
4. Trial population.
5. Duration of time participants was receiving RRT (AKI or ESKD).
6. Intervention.
7. Domain of the intervention (medication, medical device or dialysis parameter).
8. Primary outcome measure.
9. Type of primary outcome variable (continuous or dichotomous).
10. Nature of primary outcome (surrogate outcomes, patient-reported measures or hard clinical outcomes).
11. Dialysis modality (haemodialysis, peritoneal dialysis, haemodiafiltration or haemofiltration).
12. Sample size.
13. The country of the lead investigator.
14. The funder of the study (public or private).
15. Published in journal with impact factor >10.

### Assessment of the quality of the studies: risk of bias

We will use the Cochrane Risk of Bias Tool[25] to assess methodological quality of eligible trials including random sequence generation, allocation concealment, blinding of participants and healthcare personnel, blinded outcome assessment, completeness of outcome data, evidence of selective reporting and other biases. Risk of bias assessments will be performed independently by two reviewers (SC, RPPM), and disagreements were resolved by a third reviewer (CJ). If two of the domains was rated as high, the study was considered at high risk of bias. We will create a risk of bias summary table using Review Manager 5.3.[26]

### Data synthesis

A descriptive synthesis of the data will be performed. We will estimate the percentage of adaptive clinical trials in dialysis per year by dividing the adaptive clinical trials per year by the total number of RCTs per year.

### Analysis by subgroups

We will report overall outcomes and outcomes by dialysis modality (haemodialysis, peritoneal dialysis, haemodiafiltration and haemofiltration).

### Study status

This systematic review will start in April 2020.

### Patient and public involvement

There was no formal patient and public involvement in the design of this study.

### Ethics and dissemination

Ethical approval was not required for this study. We will publish the results of this systematic review in a peer-reviewed journal.

## DISCUSSION

The number of clinical trials performed in ESRD is low compared with other medical subspecialties.[27] Additionally, the cost of conducting RCTs are rising[24] and there are reduced funding sources available, especially for nephrology.[4] Adaptive clinical trials hold the potential to increase the efficiency and number of RCTs in ESRD and dialysis.

We expect to provide the following results: first, we will report on the proportion of RCTs in dialysis that include an AD; second, we will outline the most popular type of AD in ESRD trials; third, we will report the main dialysis modalities where ADs are used and possibly highlight underutilisation in additional modalities; and fourth, we will report the geographical locations and funding pattern for trials using AD in dialysis.

Furthermore, the main impact from this systematic review will be increased awareness in the nephrology community of the potential benefit of using ADs in clinical trials. Having clear examples and use cases of successful AD will stimulate further use of these designs. Patients with ESKD are often excluded from RCTs, we urgently need novel design methods for investigating treatments in this underserved population. This systematic review will highlight ADs as one part of the solution to this problem. A secondary impact from this systematic review will be an up-to-date census of all RCTs in dialysis with a yearly count of trials. This information can be used to influence funders and policymakers about the importance of funding nephrology and especially ESKD research.

### Limitations

This review will have potential limitations including publication and reporting bias. We will not be able to include studies with unpublished data and we will misclassify studies that do not have clear reporting of ADs in their methodology.

**Contributors** CJ, RPM and MOD designed the study. All authors reviewed and approved the final version of the protocol. CJ is the guarantor of the review.

**Funding** This work was performed within the Irish Clinical Academic Training (ICAT) Programme, supported by the Wellcome Trust and the Health Research Board (Grant Number 203930/B/16/Z), the Health Service Executive, National Doctors Training and Planning and the Health and Social Care, Research and Development Division, Northern Ireland. The funding source had no role in the study design, analysis or writing of report.

**Competing interests** None declared.

**Patient consent for publication** Not required.

**Provenance and peer review** Not commissioned; externally peer reviewed.

**ORCID iDs**
Conor Judge http://orcid.org/0000-0001-9473-2920
Robert P Murphy http://orcid.org/0000-0001-5446-4175

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
