## [Reviewer comments · BMJ Open]

ARTICLE DETAILS

TITLE (PROVISIONAL)	Adaptive design methods in dialysis clinical trials– a systematic review protocol
AUTHORS	Judge, Conor; Murphy, Robert P; Cormican, Sarah; Smyth, Andrew; O'Halloran, Martin; O'Donnell, Martin

VERSION 1 – REVIEW

REVIEWER	John Peipert Northwestern University, Department of Medical Social Sciences
REVIEW RETURNED	23-Jan-2020

GENERAL COMMENTS	I believe this is a well-needed review, and it seems to be very strong methodologically. However, I think more work should be done to set-up the case for the systematic review in the Introduction. Similarly, I think the authors should add to the Discussion the expected impact of the review's results. Specific points are below. - I do not think the Introduction makes the case for why adaptive trials in particular would be useful in the field of ESRD. What specific problems in previous ESRD trials would adaptive designs help to solve? What are some of the specific deficiencies of ESRD trials that could be addressed with adaptive designs? You may want to comment on how adaptive designs have been helpful in other areas of medicine and, therefore, could be useful in ESRD. The reader is left wondering why this issue is important.- In the description of Type of Outcome, the authors note that they will include “patient-centered outcomes”. Will this include only patient reported outcomes, or other types of patient centered assessments? E.g., performance and observer reports can be patient centered, though not patient reported.- The use of RobotSearch is innovative and interesting, but I am wondering if there will be any human-based “reality checks” on the results. I am not clear how well validated this machine learning approach is for the stated purpose.- Will the trials' eligibility criteria be extracted as well?- The Discussion is thin. I think you could add more on the expected impact and usefulness of the results. How do you see the results being implemented most directly?- Very small point: p. 7, lines 35-37: “a greater deal” is a bit informal or slang.
---

REVIEWER	Nithin Karakala University of Arkansas for Medical Sciences
REVIEW RETURNED	23-Jan-2020

GENERAL COMMENTS	There is an increase in the number of published adaptive studies in
---

	the recent years. The strengths of such studies include the ability of the researchers to modify the study population, intervention (like drug dosing, intervention modality, etc), and even outcomes during the study period. This gives the researches to constantly update and modify the protocol to better answer the bigger clinic picture. The greatest strength of such studies is also the greatest pitfall of such studies. These kind of studies can have a greatly variable and changing study population, changes to the methods could be made to benefit one cohort over the other, and would introduce bias. In an age where physicians are more interested in only conclusions of the abstract, it is important to put forth a strong argument as to why one should read the whole paper including the methods. The authors of this manuscript are trying to make a strong case to show us that it is important to understand the methodology of the studies in ESRD patient. What this paper is missing is the results. The manuscript can be considered a good methods section to a potentially strong paper when the authors have the results to the proposed study.
--	--

REVIEWER	Olli Saarela University of Toronto, Canada
REVIEW RETURNED	17-Mar-2020

GENERAL COMMENTS	Summary: The authors present a protocol for a systematic methodological review on the use of adaptive trial designs in dialysis trials. Search is for the co-occurrence of various dialysis and adaptive design search terms. A dialysis only search is also going to be done to identify the denominator of all dialysis trials. A machine learning algorithm is going to be used to identify trials from the search results, and the abstracts are then reviewed manually to identify adaptive trials. The study period is from database inception to beginning of 2020. Standard guidelines are going to be used for study workflow and reporting. I had the following comments. Comments:  - p. 7, l. 49-45: Maybe define here briefly what is meant by adaptive, and explain the benefits of adaptation. - p. 7. l. 58: "per 1000 clinical trials". What is characterized here looks like a proportion rather than a rate, so would be more straightforward to report it as a percentage. - p. 11, l. 35-36: "We will estimate the number of adaptive clinical trials per 1000 clinical trials in dialysis per year by dividing the adaptive clinical trials per year by the total number of randomised controlled trials per year and multiplying by 1000." Same comment as above, I don't see the need to present these numbers as rates. - Describing the time trends in the numbers and proportions is an important objective, since adaptive designs, especially Bayesian, are a relatively new alternative. Would there be any particular timepoint that would be relevant for a before/after comparison?
---

VERSION 1 – AUTHOR RESPONSE

Reviewer: 1

Reviewer Name: John Peipert

Institution and Country: Northwestern University, Department of Medical Social Sciences

Please state any competing interests or state 'None declared': None declared

Please leave your comments for the authors below

I believe this is a well-needed review, and it seems to be very strong methodologically. However, I think more work should be done to set-up the case for the systematic review in the Introduction. Similarly, I think the authors should add to the Discussion the expected impact of the review's results. Specific points are below.

- I do not think the Introduction makes the case for why adaptive trials in particular would be useful in the field of ESRD. What specific problems in previous ESRD trials would adaptive designs help to solve? What are some of the specific deficiencies of ESRD trials that could be addressed with adaptive designs? You may want to comment on how adaptive designs have been helpful in other areas of medicine and, therefore, could be useful in ESRD. The reader is left wondering why this issue is important.

Response:

Thank you for this comment. We have updated our introduction and included more information as to why adaptive design methods are an important issue in ESKD trials. We have included specific problems associated with previous ESKD trials including over reliance on observational data for trial design and incorrect extrapolation of results from other non-CKD populations. Furthermore, we have illustrated how adaptive design methods have solved these problems in other specialities. We think that the case for AD in ESKD is now stronger after your suggestions.

Changes to manuscript:

Pages 4-5, Lines 75-93.

“A specific problem in previous ESRD trials includes over reliance on observational data in the design of clinical trials [1]. Observational data are used to inform assumptions about the expected effect size and standard deviation of effect size for power calculations. If these assumptions are incorrect, they can result in an underpowered trial with an insufficient sample size to answer the research question. Adaptive sample size re-estimation has been used successfully in cardiology trials to help this problem [2]. Planned blinded sample size re-estimation can discover incorrect underlying assumptions and trigger increased recruitment targets mid trial to ensure adequate power to answer the research question. A second specific problem in previous ESRD trials is extrapolation of results from non-CKD populations. An example of this was the 4D study [3], where 20mg/day of Atorvastatin in patients with diabetes and ESRD did not reduce cardiovascular events, despite showing a 20-30% reduction in non-CKD populations [4]. One concern was that a single level of statin or a single low-density lipoprotein cut-off would not convey the same benefit in a ESKD population compared to a non-CKD population [1]. Adaptive multi-arm multi-stage (MAMS) trials have been successfully used in HIV trials, Telmisartan and Insulin Resistance in HIV (TAILoR) used a multi-arm multi-stage design with one interim analysis to access three telmisartan doses (20, 40 and 80mg daily) [5]. MAMS designs are partially suited for in ESKD populations where, compared to non-CKD populations, pharmacokinetic differences can be large and alternative drug doses are often required [6].”

- In the description of Type of Outcome, the authors note that they will include “patient-centered outcomes”. Will this include only patient reported outcomes, or other types of patient centered assessments? E.g., performance and observer reports can be patient centered, though not patient reported.

Response:

Thank you for this comment. We read your paper titled “Using Patient-reported measures in dialysis clinics” and this was very helpful [7]. For our narrative review, we wanted to summarise included studies by outcome type. We used the phrase “Patient-centred” to reflect outcomes important to dialysis patients e.g. Fatigue in the Standardised outcomes in Nephrology (SONG-HD) [8]. However, based on your comment, we have now updated our methods to “Patient-Reported measures” as this is more correct.

Changes to manuscript:

Page 8, Lines 168-169

“Nature of primary outcome (Surrogate outcomes, patient-reported measures or hard clinical outcomes).”

- The use of RobotSearch is innovative and interesting, but I am wondering if there will be any human-based “reality checks” on the results. I am not clear how well validated this machine learning approach is for the stated purpose

Response:

Thank you for this comment. RobotSearch’s machine learning algorithm was trained using the Cochrane Crowd RCT set and externally validated on the Clinical Hedges dataset for detection of randomised controlled trials across many specialities [9]. We will use the parameters to optimise the algorithm for high sensitivity, i.e. a large number of true positive and a large number of false positives. This high sensitivity search results will then be checked by two authors (10%) of the negative studies after the RobotSearch screening stage to confirm they are not RCTs.

Changes to manuscript:

Page 7, Lines 145-46

“We will manually confirm a random sample (10%) of studies classified as “Not RCT” after the RobotSearch screening step.”

Page 7, Lines 146-148

“We will then review the title and abstracts of studies to confirm that they are RCTs and identify trials with adaptive design methods for inclusion or exclusion.”

- Will the trials’ eligibility criteria be extracted as well?

Response:

Thank you for this comment. This will be a manual check by two authors in parallel (CJ and RM). Disagreement will be resolved by a third reviewer (SC).

Changes to manuscript:

Page 7, Lines 146-148

“We will then review the title and abstracts of studies to confirm that they are RCTs and identify trials with adaptive design methods for inclusion or exclusion.”

- The Discussion is thin. I think you could add more on the expected impact and usefulness of the results. How do you see the results being implemented most directly?

Response:

Thank you for this comment. We have updated the discussion section to include the expected impact and usefulness of the results.

Changes to manuscript:

Page 10, Lines 209-217

“Furthermore, the main impact from this systematic review will be increased awareness in the nephrology community of the potential benefit of using adaptive designs in clinical trials. Having clear examples and use cases of successful adaptive design will stimulate further use of these designs. ESKD patients are often excluded from randomised controlled trials, we urgently need novel design methods for investigating treatments in this underserved population. This systematic review will highlight adaptive designs as one part of the solution to this problem. A secondary impact from this systematic review will be an up to date census of all randomised controlled trials in dialysis with a yearly count of trials. This information can be used to influence funders and policy makers about the importance of funding nephrology and especially ESKD research.”

- Very small point: p. 7, lines 35-37: “a greater deal” is a bit informal or slang.

Response:

Thank you for this comment. We have updated this sentence.

Change to manuscript:

Page 4, Lines 65-66

“Additionally, people with End Stage Renal Disease experience higher mortality [10], morbidity and worse quality of life than the general public [11].”

Reviewer: 2

Reviewer Name: Nithin Karakala

Institution and Country: University of Arkansas for Medical Sciences

Please state any competing interests or state ‘None declared’: None

Please leave your comments for the authors below

There is an increase in the number of published adaptive studies in the recent years. The strengths of such studies include the ability of the researchers to modify the study population, intervention (like drug dosing, intervention modality, etc), and even outcomes during the study period. This gives the researches to constantly update and modify the protocol to better answer the bigger clinic picture. The greatest strength of such studies is also the greatest pitfall of such studies. These kind of studies can have a greatly variable and changing study population, changes to the methods could be made to benefit one cohort over the other, and would introduce bias.

In an age where physicians are more interested in only conclusions of the abstract, it is important to put forth a strong argument as to why one should read the whole paper including the methods. The authors of this manuscript are trying to make a strong case to show us that it is important to understand the methodology of the studies in ESRD patient.

What this paper is missing is the results.

The manuscript can be considered a good methods section to a potentially strong paper when the authors have the results to the proposed study.

Response:

Thank you for this comment. We agree, the most interesting part of this study will be the results. We have not commenced data extraction as our protocol (this manuscript) was still in review. We will publish our results of this study as soon as available.

Reviewer: 3

Reviewer Name: Olli Saarela

Institution and Country: University of Toronto, Canada

Please state any competing interests or state 'None declared': None declared

Please leave your comments for the authors below

Summary:

The authors present a protocol for a systematic methodological review on the use of adaptive trial designs in dialysis trials. Search is for the co-occurrence of various dialysis and adaptive design search terms. A dialysis only search is also going to be done to identify the denominator of all dialysis trials. A machine learning algorithm is going to be used to identify trials from the search results, and the abstracts are then reviewed manually to identify adaptive trials. The study period is from database inception to beginning of 2020. Standard guidelines are going to be used for study workflow and reporting. I had the following comments.

Comments:

- p. 7, l. 49-45: Maybe define here briefly what is meant by adaptive, and explain the benefits of adaptation.

Response:

Thank you for this comment. We have added details about adaptive design and the benefits of adaptation.

Changes to manuscript:

Page 4, Lines 67-72

“Adaptive clinical trials use the results from interim data analysis of ongoing trials to modify the study design in a predefined way [12]. This is performed without undermining the integrity or validity of the trial and thus preserving the Type 1 Error (False Positive) rate. The most common type of adaptive design is the Group Sequential Trial, where planned interim analysis enables stopping of trials for efficacy or futility. Other designs included sample size re-estimation, multi-arm multi-stage trials, adaptive randomisation, biomarker adaptive and seamless phase II/III trials [13].”

- p. 7. l. 58: "per 1000 clinical trials". What is characterized here looks like a proportion rather than a rate, so would be more straightforward to report it as a percentage.

Response:

Thank you for this comment. We will now report as a percentage rather than a proportion.

Changes to manuscript:

Page 2, Lines 31-32

“We will estimate the percentage of adaptive clinical trials per year in dialysis.”

Page 5, Line 98

“4) estimate the percentage of adaptive clinical trials in dialysis”

Page 9, Lines 184-186

“We will estimate the percentage of adaptive clinical trials in dialysis per year by dividing the adaptive clinical trials per year by the total number of randomised controlled trials per year.”

- p. 11, l. 35-36: "We will estimate the number of adaptive clinical trials per 1000 clinical trials in dialysis per year by dividing the adaptive clinical trials per year by the total number of randomised controlled trials per year and multiplying by 1000." Same comment as above, I don't see the need to present these numbers as rates.

Response:

Thank you for this comment. We will now report as a percentage rather than a proportion and have updated our methods to reflect this.

- Describing the time trends in the numbers and proportions is an important objective, since adaptive designs, especially Bayesian, are a relatively new alternative. Would there be any particular timepoint that would be relevant for a before/after comparison?

Response:

Thank you for this comment. We are not aware of a suitable timepoint for a before and after comparison. We reviewed previous systematic reviews that reported year of study but did not find a suitable cut-off [14]. Pre and post the 2015 publication of the “Adaptive Designs for Medical Device Clinical Studies - Guidance for Industry and Food and Drug Administration Staff” by the FDA is possibly an interesting comparison but at present, we have not included a pre-defined cut-off. We are happy to take suggestions for a particular timepoint.

VERSION 2 – REVIEW

REVIEWER	John Peipert Northwestern University, Department of Medical Social Sciences
REVIEW RETURNED	16-May-2020

GENERAL COMMENTS	The authors have done a nice job responding to each of my comments. This is a nice and helpful protocol for a study expected to add useful new knowledge to the field
---

REVIEWER	Olli Saarela University of Toronto, Canada
-----------------	---

REVIEW RETURNED	18-May-2020
-------------

GENERAL COMMENTS	No further comments.
----------------------